# Enhancement of chiral edge currents in $(d+1)$-dimensional atomic Mott-band hybrid insulators

Matteo Ferraretto[1], Andrea Richaud[1*], Lorenzo Del Re[2,3], Leonardo Fallani[4,5,6] and Massimo Capone[1,7]

**1** Scuola Internazionale Superiore di Studi Avanzati (SISSA),
Via Bonomea 265, I-34136, Trieste, Italy
**2** Max-Planck-Institute for Solid State Research, 70569 Stuttgart, Germany
**3** Department of Physics, Georgetown University, 37th and O Sts.,
NW, Washington, DC 20057, USA
**4** Department of Physics and Astronomy, University of Florence, 50019 Sesto Fiorentino, Italy
**5** LENS, European Laboratory for Non-Linear Spectroscopy, 50019 Sesto Fiorentino, Italy
**6** INO-CNR Istituto Nazionale di Ottica del CNR, Sezione di Sesto Fiorentino,
50019 Sesto Fiorentino, Italy
**7** CNR-IOM Democritos, Via Bonomea 265, I-34136 Trieste, Italy

⋆ arichaud@sissa.it

## Abstract

We consider the effect of a local interatomic repulsion on synthetic heterostructures where a discrete synthetic dimension is created by Raman processes on top of $SU(N)$-symmetric two-dimensional lattice systems. At a filling of one fermion per site, increasing the interaction strength, the system is driven towards a Mott state which is adiabatically connected to a band insulator. The chiral currents associated with the synthetic magnetic field increase all the way to the Mott transition, where they reach the maximum value, and they remain finite in the whole insulating state. The transition towards the Mott-band insulator is associated with the opening of a gap within the low-energy quasiparticle peak, while a mean-field picture is recovered deep in the insulating state.



# 1 Introduction

Current experimental platforms based on ultracold fermionic atoms trapped in $d$-dimensional optical lattices allow for the quantum simulation of the $SU(N)$-symmetric Hubbard model, where fermions have $N$ internal states with $N \geq 2$ [1]. This enhanced symmetry property derives from the vanishing electronic angular momentum in the atomic ground state of alkaline-earth atoms (like $^{87}$Sr [2]), and of some heavy Lanthanides (like $^{173}$Yb [3]), which ensures a perfect decoupling between electronic and nuclear degrees of freedom. The nuclear angular momentum therefore acts like an internal degree of freedom for the atom, providing the system with up to $N = 6$ flavors in the case of $^{173}$Yb and up to $N = 10$ flavors in the case of $^{87}$Sr.

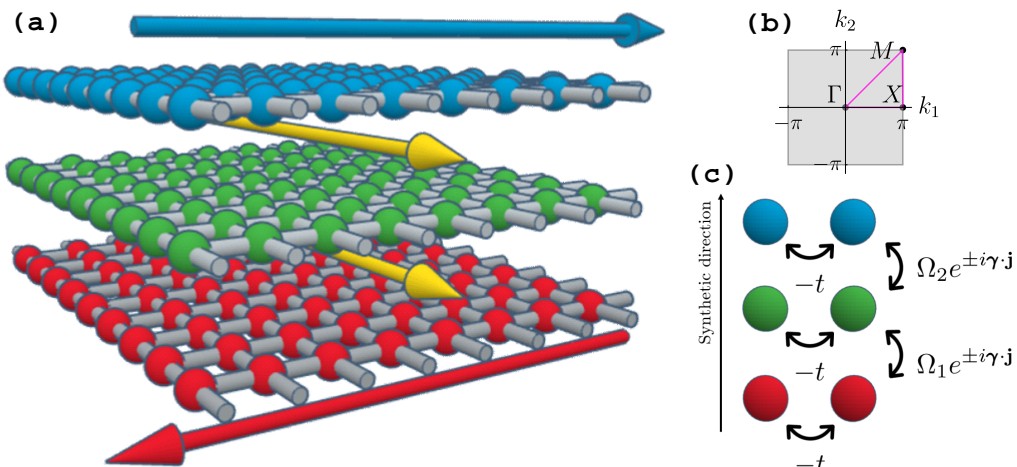

Figure 1: (a) Sketch of the $(2+1)$-dimensional synthetic heterostructure with $N = 3$ flavors. Each sphere represents a site $(\mathbf{i}, m)$ which can be occupied by a fermion. Black arrows correspond to the synthetic magnetic field $\propto \gamma$, while the blue and the red arrows represent flavor-resolved currents. (b) First Brillouin zone associated to the two dimensional square lattice and the high-symmetry path $\Gamma X M \Gamma$. (c) Illustration of real and Raman hopping processes.

The picture becomes particularly rich when the $SU(N)$ symmetry is explicitly broken, for instance by means of laser-induced Raman processes in the atoms, which amount to the absorption and the subsequent emission of polarized photons, that induce a change in the nuclear angular momentum state [4]. The Raman processes occur locally in space, but they can be effectively regarded as tunneling events between internal degrees of freedom; the latter can thus be described in terms of discretized positions along an extra *synthetic* dimension, leading to quantum simulators of $(d + 1)$-dimensional structures where $d$ is the spatial dimensionality. Suitably tuning the direction of the wave vector of the Raman laser beams, the tunneling matrix element along the synthetic dimension acquires a phase factor dependent on the real lattice site index, thus mimicking the effect of an external gauge field coupled to the system [5–7]. Moreover, the explicit $SU(N)$-breaking can lead to a flavor-selective localization of the fermions [8,9], which mimics the orbital-selective physics observed in solid-state platforms [10].

Many features of these systems have been investigated both theoretically and experimentally, mostly in the context of one-dimensional chains [7,11–19]. Following the seminal proposal of a synthetic ladder whose plaquettes are pierced by a uniform magnetic flux [7], and the experimental demonstration of chiral states localized on the synthetic edges [12], a number of remarkable effects have been pointed out. The latter include, but are not limited to, the presence of topological phases [20], Laughlin-like states [21], helical liquids [22], resonant dynamics [23] and the universality of the Hall response [24].

Conversely, the physics of these systems is essentially unexplored in dimensionalities greater than $(1 + 1)$. The $(2 + 1)$-dimensional system, for instance, is particularly interesting as it can be regarded as a synthetic heterostructure subject to strong coplanar magnetic fields and strong interlayer Coulomb repulsions (see Fig. 1). This opens the door to the quantum simulation of the basic building blocks of multilayer quantum materials [25] and to the identification of new exotic phases of matter. A deep understanding of the latter may, in fact, pave the way towards the engineering of new generations of solid-state devices [26] where one can exploit quantum mechanics to enhance functional properties like, e.g., coherent transport [27]. Further increasing the dimensionality of the real lattice, one can in principle realize quantum simulators of $(3+1)$-dimensional quantum systems [28] thus accessing novel phases [29], not accessible in three dimensions [30].

In the present work, we investigate the role of strong interatomic repulsion in the chiral properties of the system in $(2 + 1)$-dimensional structures and we compare our results with $(1 + 1)$ systems; in particular, we analyze in detail how the persistent chiral currents characteristic of the non-interacting system are modified across the interaction-driven Mott metal-insulator quantum phase transition. Our results are mainly based on dynamical mean field theory (DMFT) [31], which allows for an unbiased treatment of strong correlations for every parameter regime.

The manuscript is organized as follows: in Sec. 2 we define the model under investigation by introducing the Hamiltonian and the current operators; in Sec. 3 we show that interactions can boost chiral currents and that the latter are persistently non-zero in the insulating phase. After that, in Sec. 4, we focus on the $(2 + 1)$-dimensional lattice and analyze the role of strong particle correlations by discussing the spectral properties of the system; in Sec. 5, we provide a different perspective on the chiral currents by discussing a spin model that effectively captures the physics at strong interactions, when local density fluctuations are inhibited. In Sec. 6, we show that the discussed phenomenology is within the reach of current experimental apparatuses and, finally, Sec. 7 is devoted to concluding remarks. The description of the static and dynamical mean field approaches used throughout the work are discussed in more detail in the Appendices A and B. Finally, in Appendix C we present the exact calculation of the equilibrium current patterns of the system in small one-dimensional clusters, taking into

account the role of temperature and the presence of open boundary conditions.

## 2 The model

In this section we describe the Hamiltonian model of interacting fermions with explicitly broken $SU(N)$ symmetry induced by Raman processes on a $d$-dimensional hypercubic lattice and we introduce the definition of chiral current.

A $d$-dimensional hypercubic lattice is generated by a set of orthogonal lattice vectors $\mathbf{u}_a$ with $a = 1, ..., d$, where conventionally $|\mathbf{u}_a| = 1$. We consider a linear size $L$, hence a total number of $L^d$ sites and periodic boundary conditions (PBC) along all the spatial directions. As represented in panel (a) of Fig. 1, the internal degree of freedom (flavor) can be regarded as an extra dimension (synthetic dimension) on top of the $d$ spatial dimensions, so that the system can be effectively described in terms of spinless fermions moving on a $(d + 1)$-dimensional space, where the number of sites along the synthetic dimension is limited to $N$. Besides standard hopping processes between nearest neighbors along the real directions, we also introduce hopping processes along the synthetic direction in order to account for the effect of flavor-changing Raman processes. Such processes, that explicitly break the $SU(N)$ internal symmetry of the system, are local in real space and can be tuned to occur with a site-dependent complex amplitude, thus mimicking the presence of an external gauge field acting on the neutral atoms [7].

The system is thus described by means of the following Hamiltonian:

$$H = -t \sum_{\langle \mathbf{ij} \rangle} \sum_{m=-\mathcal{I}}^{\mathcal{I}} \left( c_{\mathbf{i},m}^\dagger c_{\mathbf{j},m} + \text{h.c.} \right) + \sum_{\mathbf{j}} \sum_{m=-\mathcal{I}}^{\mathcal{I}-1} \Omega_m \left( e^{-i\boldsymbol{\gamma} \cdot \mathbf{j}} c_{\mathbf{j},m}^\dagger c_{\mathbf{j},m+1} + \text{h.c.} \right) + \frac{U}{2} \sum_{\mathbf{j}} n_{\mathbf{j}} (n_{\mathbf{j}} - 1), \quad (1)$$

where $c_{\mathbf{i},m}$ ($c_{\mathbf{i},m}^\dagger$) annihilates (creates) a fermion with flavor index $m$ on the real lattice site labeled by the vector $\mathbf{i}$, $\mathcal{I} = (N-1)/2$, $n_{\mathbf{i}} = \sum_{m=-\mathcal{I}}^{\mathcal{I}} c_{\mathbf{i},m}^\dagger c_{\mathbf{i},m}$ is the local number operator, and conventionally $\boldsymbol{\gamma} = \gamma \mathbf{u}_1$. The symbol $\langle \mathbf{ij} \rangle$ here means that the sum runs over all the possible bonds connecting two nearest neighbor sites labeled $\mathbf{i}$ and $\mathbf{j}$, where each bond is only counted once. We assume that the Raman tunneling couples only sites that are nearest neighbors in the synthetic dimension, where the boundary conditions are open. Furthermore, we assume a uniform synthetic tunneling amplitude: $\Omega_m \equiv \Omega \, \forall m$, although in principle tunneling imbalances can be taken into account, such as in Refs. [8, 12]. The interaction term $\propto U > 0$ penalizes double and multiple occupations, it is *local* with respect to the $d$ spatial dimensions, yet it couples all the states in the synthetic dimension with a constant interaction. Unlike the Raman coupling $\propto \Omega$, this term is $SU(N)$-invariant, meaning that scattering events do not allow for flavor redistribution. Since we aim at investigating the role of the Mott physics, we work at integer filling and we consider the specific case of one fermion per site $n = L^{-d} \sum_{\mathbf{i}} \langle n_{\mathbf{i}} \rangle = 1$.

Since the complex phase of the Raman tunneling is site dependent, Hamiltonian (1) is not translation invariant along the real direction $\mathbf{u}_1$. We can restore translation invariance via the change of basis

$$c_{\mathbf{j},m} = e^{im\boldsymbol{\gamma} \cdot \mathbf{j}} d_{\mathbf{j},m}, \quad (2)$$

which leads to the transformed Hamiltonian

$$\mathcal{H} = -t \sum_{\langle \mathbf{ij} \rangle} \sum_{m=-\mathcal{I}}^{\mathcal{I}} \left( e^{im\boldsymbol{\gamma} \cdot (\mathbf{j}-\mathbf{i})} d_{\mathbf{i},m}^\dagger d_{\mathbf{j},m} + \text{h.c.} \right) + \frac{U}{2} \sum_{\mathbf{j}} n_{\mathbf{j}} (n_{\mathbf{j}} - 1) + \sum_{\mathbf{j}} \sum_{m=-\mathcal{I}}^{\mathcal{I}-1} \Omega_m \left( d_{\mathbf{j},m}^\dagger d_{\mathbf{j},m+1} + \text{h.c.} \right), \quad (3)$$

where the flavor-resolved number operator remains unchanged, i.e. $d_{\mathbf{j},m}^\dagger d_{\mathbf{j},m} = c_{\mathbf{j},m}^\dagger c_{\mathbf{j},m} = n_{\mathbf{j},m}$.

Since we are mostly interested in studying the persistent currents in the ground state of the system, we introduce the current operator for the $m$-th species along the direction $\mathbf{u}_a$

$$I_{a,m} = -\frac{it}{L^d} \sum_{\mathbf{i}} \left( e^{im\boldsymbol{\gamma}\cdot\mathbf{u}_a} d_{\mathbf{i},m}^{\dagger} d_{\mathbf{i}+\mathbf{u}_a,m} - \text{h.c.} \right), \tag{4}$$

and the full current vector $\mathbf{I}_m = \sum_a I_{a,m} \mathbf{u}_a$ [32–34]. Switching to the momentum representation of the fermionic operators $d_{\mathbf{k},m} = L^{-d/2} \sum_{\mathbf{j}} e^{i\mathbf{k}\cdot\mathbf{j}} d_{\mathbf{j},m}$, we can recast Eq. (4) as

$$I_{a,m} = \frac{2t}{L^d} \sum_{\mathbf{k}} \sin(k_a + m\gamma_a) \, n_{\mathbf{k},m}, \tag{5}$$

where $k_a = \mathbf{k} \cdot \mathbf{u}_a$ and $\gamma_a = \boldsymbol{\gamma} \cdot \mathbf{u}_a$ [35]. We can also introduce the current operator along the rung of the ladder or heterostructure as

$$I_{\mathbf{i}} = i\Omega \sum_{m=-\mathcal{I}}^{\mathcal{I}-1} \left( d_{\mathbf{i},m}^{\dagger} d_{\mathbf{i},m+1} - \text{h.c.} \right). \tag{6}$$

Remarkably, as discussed in more detail in Appendix C, in any stationary state, the current pattern satisfies the equivalent of the Kirchhoff's current law for every single node $(\mathbf{i}, m)$ of the structure.

Finally, we define the chiral current as the expectation value of the difference between the two outermost flavor currents:

$$\mathbf{I}_{\text{chir}} = \langle \mathbf{I}_{-\mathcal{I}} - \mathbf{I}_{\mathcal{I}} \rangle. \tag{7}$$

Interestingly, the chiral current vector can be written in a compact form as the expectation value of the gradient of Hamiltonian (3) with respect to the synthetic flux vector $\boldsymbol{\gamma}$, up to a suitable normalization:

$$\mathbf{I}_{\text{chir}} = -\frac{2L^{-d}}{N-1} \langle \nabla_{\boldsymbol{\gamma}} \mathcal{H} \rangle. \tag{8}$$

Before moving to the discussion of the particular cases $N = 2$ and $N = 3$, it is worth mentioning that for an arbitrary $N$, Hamiltonian (3) has a point reflection symmetry, as long as the Raman tunneling amplitudes do not depend on $m$. This means that the Hamiltonian is invariant under a simultaneous reflection on the real and the synthetic space: i.e. $c_{\mathbf{i},m} \to c_{-\mathbf{i},-m}$. As a consequence of this symmetry, it is possible to prove that $\langle I_{a,-m} \rangle = -\langle I_{a,m} \rangle$, which in turn implies that, when $N$ is odd and $m = 0$ is a possible flavor index, the current on the central leg vanishes: $\langle I_{a,0} \rangle = 0$. However we remark that in general the currents flowing on inner legs (i.e. legs with $m \neq 0, \pm\mathcal{I}$) are non-vanishing, so that when $N > 3$ one can in principle define $N/2$ chiral currents if $N$ is even, and $(N-1)/2$ chiral currents if $N$ is odd.

# 3 Chiral currents

We start by discussing the properties of the non-interacting system ($U = 0$) in $(2+1)$-dimensions. In this case, Hamiltonian (3) can be easily diagonalized and the resulting band diagram is illustrated along a typical high-symmetry path of the first Brillouin zone in Fig. 1 (b).

At unitary filling we find a metal-insulator transition as a function of $\Omega/t$. As shown in Fig. 2 for $N = 2$ (first row) and $N = 3$ (second row), for small $\Omega/t$ we find a metallic state with the Fermi energy crossing at least two bands, while for large values of $\Omega/t$ a band gap opens up. Interestingly, increasing $\Omega/t$ changes substantially the flavor character of the single-particle states, that we represent through a colour scale [7]. As we can see in Fig. 2, for

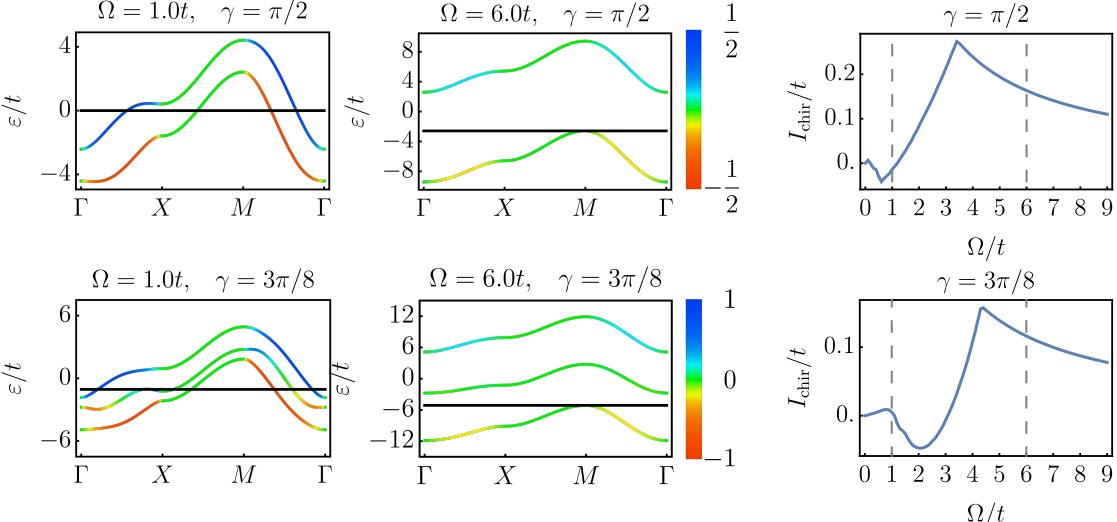

Figure 2: Band structure of the 2-flavor (top) and 3-flavor system (bottom) along the high symmetry path of the Brillouin zone. The color pattern reflects the flavor polarization of the corresponding state, while the black horizontal line shows the highest occupied energy level. The chiral currents depend on the filling of single particle states and the associated polarization (see Sec. 3). The right panel shows the behavior of the chiral current across the transition between the metal and the band insulator (at the transition the chiral current has a non differentiable peak). Dashed vertical lines are values of $\Omega$ representative of the two phases for which the band structure is displayed.

small $\Omega/t$ the different bands have different weight for different flavors, while the bands for large $\Omega/t$ display smaller differences between different bands and a more homogeneous flavor content as we move in the Brillouin zone. This, in turn, reflects on the magnitude and sign of the overall chiral current (7) in the system, displayed in the right column of Fig. 2. We assume $\boldsymbol{\gamma} = \gamma \mathbf{u}_1$, so that the currents flow along the direction $\mathbf{u}_1$ and we only consider the corresponding component $I_{\text{chir}} := \mathbf{I}_{\text{chir}} \cdot \mathbf{u}_1$. Interestingly, $I_{\text{chir}}$ features a cusp-like maximum exactly at the value $\Omega/t$ at which the system undergoes the metal to band-insulator quantum phase transition.

This peculiar behavior can be simply explained in terms of the band structure. Especially when $\Omega/t$ is small, each band has a definite chirality, meaning that states with $k_1 > 0$ are typically polarized towards one external flavor, while states with $k_1 < 0$ are polarized towards the other one; however, different bands can have opposite chirality, as we see in our representative results of Fig. 2. For very small $\Omega/t$, however, the fermions populate different bands, thus there are fermions with the same lattice momentum $\mathbf{k}$ and with opposite flavor polarization. These fermions with opposite (although large) chirality give disruptive interfering contributions to the overall chiral current. When we increase $\Omega/t$, these bands are split in energy, and they are populated unevenly. Hence the chiral current increases, as we see in both plots, up to the value where we reach the transition and we enter in the insulating state.

On the other hand, increasing $\Omega/t$ reduces the flavor polarization in each state. This effect becomes dominant in the insulator, where we populate only one band, which explains the decreasing behavior of $I_{\text{chir}}$ in the insulator, where all the fermions populate the same band and have the same chirality. A simple analytical expression for the chiral current deep in the band-insulating regime ($\Omega \gg t$) can be obtained by computing the eigenstates of Hamiltonian (3) and by plugging the expectation values of $n_{\mathbf{k},m}$ into Eq. (5): we find $I_{\text{chir}} \sim \frac{t^2}{\Omega} \sin \gamma$ for

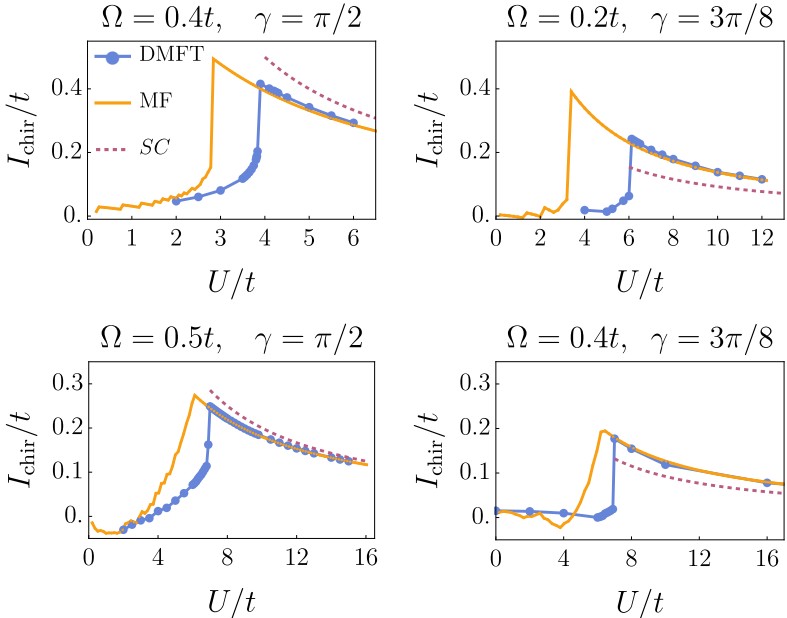

Figure 3: Chiral current as a function of the interaction $U/t$ for a $(1+1)$-dimensional structure (first row) and a $(2+1)$-dimensional structure (second row); both for $N = 2$ (left column) and $N = 3$ (right column). The curves represent results obtained with different methods: a static mean field approach (MF), dynamical mean field theory (DMFT) and an effective strong-coupling limit (SC). All the techniques confirm the presence of a non differentiable peak in the function $I_{\text{chir}}(U)$ at the transition and a $1/U$ tail in the insulating phase.

$N = 2$ and $I_{\text{chir}} \sim \frac{t^2}{\Omega} \sin\gamma(1 + 3\cos\gamma)/\sqrt{8}$ for $N = 3$, highlighting that in both cases $I_{\text{chir}} \sim \frac{1}{\Omega}$.

Naively, one would expect the inclusion of the interaction $U$ to reduce the currents and a vanishing current in the Mott insulating state. We show, instead, that this is not the case, as chiral currents can actually be *boosted* by interactions, and persist deep inside the insulating phase.

We solve the interacting problem comparing static Hartree-Fock mean-field (MF) and dynamical mean-field theories. Within a standard MF picture, the interaction simply modifies the non-interacting energy bands (Appendix A) leading to an effective band picture. On the other hand, DMFT (Appendix B) is a non perturbative approach which is reliable both at weak and at strong coupling, thus being an unbiased technique, suitable for exploring a wide range of values of $U$. Within DMFT the lattice model is mapped onto an effective impurity model that we solve at zero temperature ($T = 0$) using an exact diagonalization algorithm [36] that has been generalised to treat SU(N) Hubbard models relevant to cold atom systems. Within DMFT, the effect of the interactions is included in a flavor-dependent self-energy which retains the full frequency dependence while the momentum dependence is neglected [$\Sigma(\mathbf{k}, i\omega_n) \approx \Sigma(i\omega_n)$], as opposed to static mean field where the frequency dependence is also absent.

In Fig. 3 we show $I_{\text{chir}}$ as a function of $U$, for $N = 2$ and $N = 3$ flavors (left and right column respectively), as obtained both within MF and DMFT. We compare the $d = 2$ results with those in $d = 1$, for which our results reproduce the trends of previous calculations [16]. All the curves feature a relatively smooth growth for small $U$, interrupted by a cusp-like maximum at a critical value of $U/t$, where the system undergoes the $U$-driven metal-insulator transition, followed by a $\sim 1/U$ behavior in the insulating phase. Even more surprisingly, we observe that the chiral current is typically larger in the insulating phase than in the metallic phase, and it is

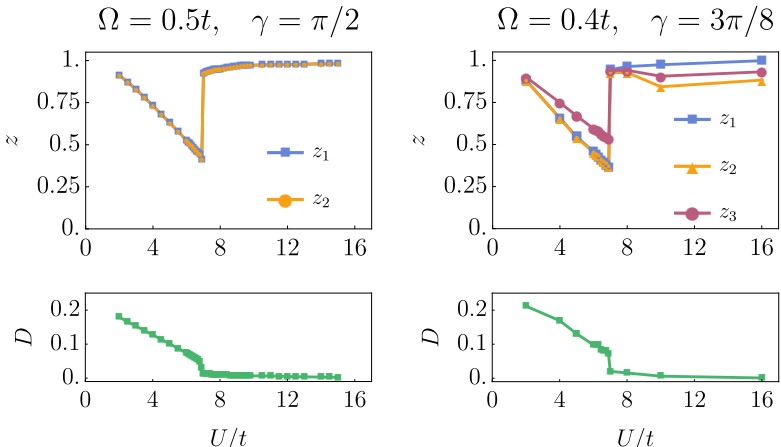

Figure 4: Upper panels: quasiparticle weights $z_\alpha$, where $\alpha = 1, ..., N$, as a function of $U$ in the $(2+1)$-dimensional system for $N = 2$ (left) and $N = 3$ (right). Lower panels: double occupancy for $N = 2$ (left) and $N = 3$ (right). Both quantities are discontinuous at the metal-insulator transition. The fact that $D$ goes to zero while $z$ remains finite ($\approx 1$) for $U > U_c$ is the hallmark of the hybrid character of the insulating phase (featuring both Mott-like and band-like properties).

maximized at the metal-insulator transition, similarly to what we found in the non-interacting limit, where the transition (in that case driven by $\Omega$) has a more conventional band character.

Remarkably, MF and DMFT provide similar qualitative results and a striking quantitative agreement at strong coupling. While for $d = 1$ the MF results underestimate the critical interaction strength (which in turn controls the maximum value of $I_{\text{chir}}$) with respect to DMFT, the agreement becomes much closer in $d = 2$. In the next section we discuss the origin of this scenario in more detail.

## 4 The metal-insulator transition

In this section we analyze the evolution of the correlation properties across the metal-insulator transition. A useful quantity to measure the degree of correlation of a system is the flavor-dependent quasiparticle weight

$$z_\alpha = \left(1 - \left.\frac{\partial \Sigma_{\alpha\alpha}(i\omega_n)}{\partial i\omega_n}\right|_{i\omega_n \to 0}\right)^{-1}, \quad (\alpha = 1, \dots, N), \tag{9}$$

which corresponds to the amplitude of the low-energy spectral weight with metallic character. A non-interacting system has $z_\alpha = 1$, while a vanishing $z_\alpha$ corresponds to the total loss of low-energy coherent spectral weight characteristic of a Mott insulator and intermediate values correspond to increasingly bad metals. In the following we compare the evolution of this quantity with the ground-state double occupancy $D = \langle L^{-d} \sum_{\mathbf{i}} \sum_{m < m'} n_{\mathbf{i},m} n_{\mathbf{i},m'} \rangle$.

Finally, we monitor the momentum-resolved single-particle spectral function which can be measured by angle resolved photoemission spectroscopy (ARPES) [37] and its cold-atom counterparts [38–40]

$$A(\mathbf{k}, \omega) = -\frac{1}{\pi} \lim_{\eta \to 0^+} \sum_{\alpha=1}^{N} \text{Im}\left[G_{\alpha\alpha}(\mathbf{k}, \omega + i\eta)\right], \tag{10}$$

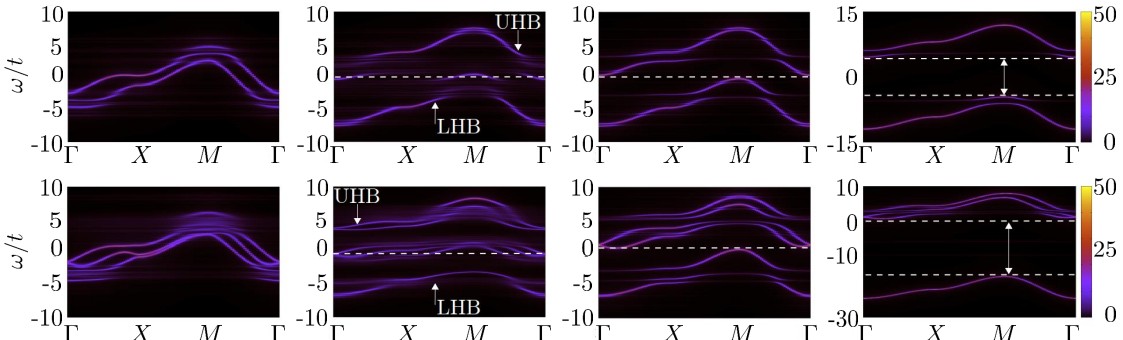

Figure 5: Evolution of the momentum-resolved spectral function obtained with DMFT upon increasing $U$ and crossing the phase transition. First (second) row corresponds to $N = 2$ and $\Omega/t = 0.5$ ($N = 3$ and $\Omega/t = 0.4$). From left to right, each panel corresponds to $U \ll U_c$, $U \rightarrow U_c^-$, $U \rightarrow U_c^+$ and $U \gg U_c$ respectively. The specific values are $U/t = 2, 6.9, 7, 16$ in the first row, and $U/t = 2, 6.9, 7, 24$ in the second. UHB (LHB) in the second column from the left indicates the (precursor of the) Upper Hubbard Band (Lower Hubbard Band).

where $G_{\alpha\alpha}(\mathbf{k}, \omega + i\eta)$ is the retarded interacting Green function for the flavor $\alpha$. The density plot of $A(\mathbf{k}, \omega)$ along the high symmetry path of the first Brillouin zone (Fig. 5) results in a generalization of the band diagram shown in Fig. 2 to the case of an interacting system.

As shown in the upper panel of Fig. 4, all $z_\alpha$ for any $\alpha$ decrease as a function of $U$ as long as we are in the metallic side of the transition. This is a signature of the increased degree of correlation of the metal and of the interaction-induced shrinking of the bands. At the same time, the double occupations decrease due to their increased energetic cost. For a standard Hubbard model without symmetry breaking, this behavior is extended all the way to the Mott transition, where the quasiparticle weight vanishes continuously. In our model we find, instead, a distinct scenario, where the quasiparticle weight remains well different from zero for any value of $U$ and it jumps to a large value close to 1 when the insulating state is reached. This reflects the fact that the self-energy becomes frequency-independent at low frequency and it is again completely different from the typical behavior of a Mott insulator, where the self-energy diverges as $1/i\omega_n$ and the quasiparticle weight vanishes. Yet, double occupancies drop to a small value at the transition, signaling that the Mott localization, which is associated with a sharp reduction of doubly occupied sites, is still taking place.

This scenario reflects on the mechanism of gap opening shown in Fig. 5. While in the symmetric Hubbard model the quasiparticle peak at the Fermi level disappears at the Mott transition leaving a preformed gap of order $U$, in the $SU(N)$-broken systems a rather large quasiparticle peak survives just before the transition, and the insulating gap arises from a splitting of such peak into two features. As a consequence the gap is not proportional to the Hubbard $U$. At the same time, analogously to the standard scenario, spectral weight moves towards high-energy features separated by an energy $U$ already in the metallic state, which are usually referred to as precursors of the Hubbard bands. Upon increasing $U$, the band gap increases, and the central spectral features (where the band gap has opened), are continuously pushed towards the preformed Hubbard bands, until they finally merge at very large $U$. Interestingly, in the latter regime where the bands are merged, the self-energy is nearly constant as a function of frequency, so $A(\mathbf{k}, \omega)$ is correctly predicted, both qualitatively and quantitatively, by a static mean-field approach (see Appendix A), consistently with the observed agreement of the chiral currents shown in Fig. 3.

As a matter of fact, the interaction drives the system towards a state which can be safely

considered a Mott state since it is stabilized by a strong suppression of doubly occupied sites, but, at the same time, is similar to a band-insulator and it can be described by static mean-field.

From an intuitive point of view, the key point is that in our system with broken $SU(N)$ symmetry there is no competition between the states selected by the Hubbard $U$, namely any state with one fermion on every site, and those favoured by the symmetry-breaking field, which are specific single-fermion states obtained as linear combinations of the different components. As a result, increasing $U$ favors the stabilization of the band insulator by reducing the weight of states with more than one fermion per site. In other words, the Mott localization and the formation of the band insulator are not competitive effects and they can actually cooperate to stabilize the same insulating state. As a matter of fact, our interaction-driven transition is very similar to the band-insulator transition that we have found and discussed in the non-interacting system.

The picture above closely resembles the insulating phase reported in Ref. [41], that the authors described as a Mott insulator *disguised* as a conventional band insulator. This is a sort of hybrid between a Mott insulator, characterized by the suppression of local density fluctuations (see lower panels of Fig. 4) and by the presence of preformed Hubbard bands, and a conventional band insulator, characterized by a frequency-independent self-energy and an effective non-interacting description. This Mott-band insulator is indeed adiabatically connected with the non-interacting band insulator discussed in Sec. 3.

All these observations about the mixed nature of the insulating state explain why a static Hartree-Fock mean-field approach agrees so well with DMFT in the insulating phase. Therefore, the arguments that we used to understand the origin of the chiral current peak at the transition can be extended to the interacting case. This result has been found also in $d = 1$ [16], where it was argued that the result was expected to hold also in higher dimensionality.

We emphasize that the close similarity between DMFT and Hartree Fock is limited to the region $U \gg U_c$; whereas the spectral properties at intermediate coupling are highly non-trivial and far from a mean-field picture, hence they require a full dynamical description to be accurately described. In particular, the opening of the gap within the quasiparticle peak is an unambiguous signature of non-trivial correlation effects which are not accessible within static mean-field.

However, as far as the evolution of the effective band structure with $U$ is concerned, we recover the same main trends of the transition we found for $U = 0$ as a function of $\Omega/t$. Therefore the arguments that we discussed in Sec. 3 in order to explain the existence of a maximum of the chiral current at the metal-insulator transition are expected to approximately hold, especially in the insulating region.

# 5 Effective strong-coupling model

In this section we discuss the strong-coupling (SC) limit $U \gg t, \Omega$ (again at unitary filling $\langle n_{\mathbf{j}} \rangle = 1$), which allows to understand the $1/U$ behavior of the chiral current. In this regime, where charge fluctuations are strongly suppressed, the low-energy properties of the system can be described by using only the flavor degrees of freedom and working in a reduced Hilbert space, characterized by Fock states with one fermion per site. The resulting Hilbert space is the tensor product over all the lattice sites of local $N$-dimensional spaces, where the canonical basis vectors represent the $N$ possible local flavor states. Analogously to the standard Hubbard model, Hamiltonian (1) maps into an effective Heisenberg-like Hamiltonian with broken

$SU(N)$ symmetry [3, 42]:

$$H^{\text{eff}} = J \sum_{\langle \mathbf{ij} \rangle} \sum_{mn} S_{\mathbf{i};m,n} S_{\mathbf{j};n,m} + \Omega \sum_{\mathbf{j}} \sum_{m=-\mathcal{I}}^{\mathcal{I}-1} \left( e^{-i\boldsymbol{\gamma}\cdot\mathbf{j}} S_{\mathbf{j};m,m+1} + \text{h.c.} \right), \tag{11}$$

where $J = 2t^2/U$ is the effective super-exchange interaction, $S_{\mathbf{j};m,n} = c_{\mathbf{j};m}^{\dagger} c_{\mathbf{j};n}$ is the ladder operator that changes the fermionic flavor at site $\mathbf{j}$ from $n$ to $m$ and satisfies the commutation relations

$$\left[ S_{\mathbf{i};m,n}, S_{\mathbf{j};p,q} \right] = \delta_{\mathbf{ij}} \left( \delta_{n,p} S_{\mathbf{i};m,q} - \delta_{m,q} S_{\mathbf{i};p,n} \right). \tag{12}$$

Only $N^2 - 1$ out of the $N^2$ ladder operators are independent, since they must satisfy the constraint $\sum_n S_{\mathbf{i},nn} = n_{\mathbf{i}} = 1$, so they can be regarded as the $N$-dimensional representation of $SU(N)$ [42].

An analogous mapping holds also for the current operators on the bonds, which can thus be written as

$$\mathbf{I}_{\mathbf{i},\mathbf{i}+\mathbf{u}_a;m}^{\text{eff}} = -iJ \sum_a \sum_n \left( S_{\mathbf{i};m,n} S_{\mathbf{i}+\mathbf{u}_a;n,m} - S_{\mathbf{i}+\mathbf{u}_a;m,n} S_{\mathbf{i};n,m} \right) \mathbf{u}_a, \tag{13}$$

and

$$I_{\mathbf{j};m,m+1}^{\text{eff}} = -i\Omega e^{-i\boldsymbol{\gamma}\cdot\mathbf{j}} S_{\mathbf{j};m,m+1} + \text{h.c.} \tag{14}$$

Interestingly, while the *total* current operator vanishes $\sum_m \mathbf{I}_{\mathbf{i},\mathbf{j};m}^{\text{eff}} = \mathbf{0}$ for any pair of neighboring sites, as one expects for an insulator, the *flavor-resolved* currents $\mathbf{I}_{\mathbf{i},\mathbf{j},m}^{\text{eff}}$ do not vanish. Most importantly, the chiral current operator is non zero and it is given by

$$\mathbf{I}_{\text{chir}}^{\text{eff}} = \mathbf{I}_{-\mathcal{I}}^{\text{eff}} - \mathbf{I}_{+\mathcal{I}}^{\text{eff}}, \tag{15}$$

where $\mathbf{I}_m^{\text{eff}} = L^{-d} \sum_{\langle \mathbf{ij} \rangle} \mathbf{I}_{\mathbf{i},\mathbf{j};m}^{\text{eff}}$.

## 5.1 Two flavors

For $N = 2$ flavors, Hamiltonian (11) reads

$$H^{\text{eff}} = 2J \sum_{\langle \mathbf{ij} \rangle} \vec{S}_{\mathbf{i}} \cdot \vec{S}_{\mathbf{j}} - \sum_{\mathbf{j}} \vec{B}_{\mathbf{j}} \cdot \vec{S}_{\mathbf{j}}, \tag{16}$$

where $\vec{B}_{\mathbf{j}} = 2\Omega(-\cos(\boldsymbol{\gamma}\cdot\mathbf{j}), \sin(\boldsymbol{\gamma}\cdot\mathbf{j}), 0)$ is an effective site-dependent magnetic field and we have introduced the standard spin operators, whose components are $S_{\mathbf{j}}^{\alpha} = \frac{1}{2} \sum_{mn} \sigma_{mn}^{\alpha} c_{\mathbf{j},m}^{\dagger} c_{\mathbf{j},n}$, $\sigma^{\alpha}$ being the $\alpha$-th Pauli matrix. Accordingly, the aforementioned Kirchhoff's current law [see Eq. (C.1) in Appendix C], is readily rephrased as

$$-2J \sum_a \left( \vec{S}_{\mathbf{j}} \times \vec{S}_{\mathbf{j}+\mathbf{u}_a} - \vec{S}_{\mathbf{j}-\mathbf{u}_a} \times \vec{S}_{\mathbf{j}} \right) + \left( \vec{S}_{\mathbf{j}} \times \vec{B}_{\mathbf{j}} \right) = \vec{0}, \tag{17}$$

which has the form of a mechanical-equilibrium condition for the effective spins. The terms in Eq. (17) have a non-trivial physical interpretation, as the chiral-current operator (15) and the synthetic dimension-current operator (14) can indeed be written as

$$\mathbf{I}_{\text{chir}}^{\text{eff}} = 4J \sum_a \sum_{\mathbf{j}} \left( \vec{S}_{\mathbf{j}} \times \vec{S}_{\mathbf{j}+\mathbf{u}_a} \right)_z \mathbf{u}_a, \tag{18}$$

$$I_{\mathbf{j};-\frac{1}{2},+\frac{1}{2}}^{\text{eff}} = \left( \vec{S}_{\mathbf{j}} \times \vec{B}_{\mathbf{j}} \right)_z, \tag{19}$$

two quantities which are proportional to the $z$-component of the *torque* exerted on the spin at site $\mathbf{j}$ by the nearest-neighbor spins and by the external magnetic field, respectively.

For large values of $U/t$, one enters the regime where $\Omega \gg J$ and thus the spins $\vec{S}_{\mathbf{j}}$ tend to align to the local effective magnetic field $\vec{B}_{\mathbf{j}}$ in spite of the spin-spin superexchange interaction $\propto J$. Thus, the cross product in Eq. (18) saturates to the value $(\sin \gamma)/4$, and so the overall chiral current goes as $2t^2 (\sin \gamma)/U$. This observation explains why $I_{\text{chir}} \propto t/U$ for large $U$. The quantitative comparison in the left panels of Fig. 3 shows that the agreement with DMFT results is remarkable in the whole insulating range.

## 5.2 Three flavors

For $N > 2$, Hamiltonian (11) can be rewritten in terms of a $N$-dimensional representation of $SU(2)$ [i.e. in terms of spin-$s$ operators, where $s = (N-1)/2$] instead of the above $N$-dimensional representation of $SU(N)$, as discussed in Ref. [43]. However, in terms of these operators, the Hamiltonian features higher-order exchange processes on top of the standard Heisenberg interaction. In the light of this, the $N = 3$ version of Hamiltonian (11) can thus be mapped into

$$H^{\text{eff}} = J \sum_{\langle \mathbf{ij} \rangle} \left[ \vec{\Sigma}_{\mathbf{i}} \cdot \vec{\Sigma}_{\mathbf{j}} + \left( \vec{\Sigma}_{\mathbf{i}} \cdot \vec{\Sigma}_{\mathbf{j}} \right)^2 \right] - \sum_{\mathbf{j}} \frac{\vec{B}_{\mathbf{j}}}{\sqrt{2}} \cdot \vec{\Sigma}_{\mathbf{j}}, \tag{20}$$

where the effective spin-1 operators are defined as

$$\Sigma_{\mathbf{j}}^x = \frac{1}{\sqrt{2}} \left( c_{\mathbf{j},0}^\dagger c_{\mathbf{j},1} + c_{\mathbf{j},-1}^\dagger c_{\mathbf{j},0} + \text{h.c.} \right),$$

$$\Sigma_{\mathbf{j}}^y = \frac{i}{\sqrt{2}} \left( c_{\mathbf{j},0}^\dagger c_{\mathbf{j},1} + c_{\mathbf{j},-1}^\dagger c_{\mathbf{j},0} - \text{h.c.} \right),$$

$$\Sigma_{\mathbf{j}}^z = n_{\mathbf{j},1} - n_{\mathbf{j},-1}, \tag{21}$$

and they satisfy the $SU(2)$ algebra $[\Sigma_{\mathbf{j}}^\alpha, \Sigma_{\mathbf{j}}^\beta] = i \varepsilon_{\alpha\beta\gamma} \Sigma_{\mathbf{j}}^\gamma$.

As we have anticipated, Hamiltonian (20) differs from (16) not only because it includes spin-1 operators, but also for the presence of a quartic interaction term. In general, it is possible to verify that every ladder operator of the $SU(3)$ representation can be written as a quadratic combination of the $SU(2)$ generators introduced in (21), and therefore we are able to express every local operator in terms of the latter. So not only the Hamiltonian, but also the current operators can now be expressed in terms of the spin-1 operators. Remarkably, the chiral current is formally equal (up to a multiplicative constant) to the one in Eq. (18), i.e.

$$\mathbf{I}_{\text{chir}}^{\text{eff}} = \frac{J}{2} \sum_a \sum_{\mathbf{j}} \left( \vec{\Sigma}_{\mathbf{j}} \times \vec{\Sigma}_{\mathbf{j}+\mathbf{u}_a} \right)_z \mathbf{u}_a, \tag{22}$$

and the synthetic-dimension current $I_{\mathbf{j};-1,0}^{\text{eff}} + I_{\mathbf{j};0,+1}^{\text{eff}} = (\vec{\Sigma}_{\mathbf{j}} \times \vec{B}_{\mathbf{j}})_z$ to the one in Eq. (19). In the limit $\Omega \gg J$, the spins $\vec{\Sigma}_{\mathbf{j}}$ tend to align to the local magnetic field $\vec{B}_{\mathbf{j}}$ and, similarly to what discussed in Sec. 5.1, the chiral current turns out to be $I_{\text{chir}} \sim t^2 (\sin \gamma)/U$ (see the right panels of Fig. 3 for a quantitative comparison). For the same reason, as discussed in Appendix C, all the synthetic-dimension currents (other than the two outer ones) in the right panel of Fig. 10 are vanishing.

## 6 Experimental realization

The experimental realization of the proposal is based on the combination of the techniques introduced in Ref. [12], where chiral currents in fermionic synthetic ladders were first measured, with those demonstrated in Ref. [8], where the Mott transition in the presence of a

coherent coupling breaking the $SU(N)$ symmetry [9] was observed. In those works, based on $^{173}$Yb fermionic atoms trapped in optical lattices, the synthetic hopping was induced by Raman transitions between a subset of states in the nuclear-spin manifold using the $^3P_1$ state as intermediate level. The value of $\gamma$ can be adjusted by controlling the angle between the two Raman beams in such a way to span the whole $[0, 2\pi]$ range.

The preparation of $SU(N)$ Fermi-Hubbard systems with unit filling can be done with conventional techniques based on the control of the atomic density and on optical potential shaping. Adiabatic state preparation can be performed by first trapping a flavor-polarized sample of atoms in the optical lattice (i.e. in a band insulating state) and then activating the synthetic tunnelling by applying a frequency sweep of the Raman coupling to bring it from being far-detuned to being resonant at the end of the preparation sequence. We note that in typical experimental realizations a weak external harmonic potential $H_{\text{trap}} = \sum_{\mathbf{i}} \sum_m w|\mathbf{i} - \mathbf{0}|^2 n_{\mathbf{i},m}$ is present. Although techniques based on arbitrary optical potentials can be used to produce flat box-like traps, we note here that the harmonic confinement is not expected to alter the main results presented in this work, for instance the $1/U$ behaviour of $I_{\text{chir}}$ in the strongly interacting regime. The reason is that, in such regime, double occupations are inhibited everywhere in the system (see Sec. 4) and cannot be unlocked by confining potentials if the latter are weak enough. Conversely, the harmonic trapping helps in making the experimental realization of the unit-filling condition robust against atom-number fluctuations.

In Appendix C we have highlighted the critical role of thermal fluctuations, leading to a reduction of the chiral currents. Recently, temperatures on the order of $0.1t/k_B$ have been reported for a $SU(N)$ Hubbard system [44] and the Raman coupling technique has already been shown to cause minimal heating, also on the order of $0.1t/k_B$ on a timescale of several tens of milliseconds in the strongly interacting regime [8], allowing for the observation of pure quantum many-body dynamics. These experimental achievements, combined with the sizable value of the chiral currents calculated under optimal conditions (at the Mott critical point), of the same order of those measured in Ref. [12], make the experimental observation of the effects proposed in this work at reach.

The spatial-resolved current patterns identified in Appendix C could be detected by imaging the system with a spin-resolved quantum gas microscope [45] after a quench in the optical lattice depth. As the hopping rate $t$ is suddenly quenched to zero, the lattice sites in the real directions are effectively decoupled and the internal state of the atoms is left free to evolve according to the Rabi coupling [last term of Eq. (1)] only. By monitoring the evolution of the local spin populations for times smaller than $\Omega^{-1}$ it is possible to extract information on the strength and sign of the rung currents before the quench. At longer times the currents will acquire an AC character, corresponding to Rabi oscillations in the spin populations (we note that Kirchoff's law mentioned in Appendix C will not hold in this out-of-equilibrium case, as the expectation value of $\dot{n}_{\mathbf{i},m}$ need not to be zero).

The momentum-resolved single-particle spectral function discussed in Sec. 4 can be measured by spectroscopic techniques based on the excitation towards non-interacting states. In the physical system considered in this work, the ARPES technique demonstrated in Ref. [39] cannot be directly implemented, as the $SU(N)$ symmetry of two-electron atoms prevents the existence of spin states in the ground-state manifold with vanishing interactions. Different techniques could be employed, for instance based on Bragg excitations towards higher lattice bands, where atoms are trapped more weakly and interaction effects are weaker (as explored for a Bose-Hubbard system in Ref. [46]), or on the excitation towards the metastable clock state $^3P_0$, where the final trapping configuration can be tailored by using state-dependent lattices, even at the tune-out magic wavelength where atoms in the $^3P_0$ state are not confined at all and would be immediately ejected from the trap.

# 7 Concluding remarks

In this work we have investigated the surprising effect of the Mott transition in synthetic ladders and heterostructures pierced by artificial magnetic fields. These systems can be realized by means of cold-atom platforms, where the presence of $N$ internal states can be mapped into a synthetic dimension, leading to $N$ legs of the resulting ladders if the spatial dimensionality $d$ is one, or $N$ layers in an effective heterostructure if $d = 2$. While in the well-known case of ladders the magnetic field is perpendicular to the ladder plane, in the case of heterostructures it is possible to simulate the presence of a strong magnetic field with no components perpendicular to the system itself.

We have focused, in particular, on the study of the chiral current, an experimentally-observable current which is experimentally accessible in cold-atom platforms, with particles flowing along the outermost legs (planes) of the synthetic ladder (heterostructure) in opposite directions. We have shown that, in the metallic phase, unexpectedly, the interparticle repulsion can *boost* the flow of counter-propagating flavor currents, which feature a sharp maximum exactly at the metal-insulator quantum phase transition. Furthermore, in the insulating phase, the chiral current is far from being suppressed; instead it fades as $1/U$ upon increasing the interaction. We have shown that the discussed results are robust against temperature variations typical of state-of-the-art experimental setups [12].

Rather surprisingly, we have proved that quantum dynamical fluctuations are suppressed in the strong-coupling regime, due to the hybrid Mott-band nature of the insulating phase; thus making it possible to interpret the system in terms of non-interacting quasiparticles populating bands renormalized by the interaction. In the metallic regime, where different bands are occupied, these particles populate states with a large, but often opposite, flavor polarization, giving interfering contributions to the resulting chiral current. On the other hand, in the insulating regime, where only one band is occupied, the particles populate states with a small, but coherent, flavor polarization, enhancing the total current. Nevertheless, the quantum dynamical fluctuations remain important in the intermediate-coupling regime, close to the phase transition.

We have further characterized the slow decrease of current in the insulating phase by means of a strong-coupling limit which maps the model onto effective interacting spins subject to an external local magnetic field, and currents as torques acting on such spins. We have shown that, for large interactions, the slow vanishing of the chiral current is directly related to the freezing of spins in the direction of the local field. Finally, we have complemented our theoretical study with a detailed experimental proposal which corroborates the possibility of observing the discussed phenomena in state-of-the-art apparatuses.

Possible future developments of the presented analysis include, but are not limited to, the study of the impact of Coulomb repulsion on the edge states of $(1 + 1)$-dimensional systems [20], the post-quench out-of equilibrium dynamics of rung and leg currents, the quantification of the flavor polarization and that of the (de)localization of the particles across the transition by means of suitable entropy-like indicators [47], the impact of a different (real) lattice geometry (e.g. honeycomb or triangular lattices [48, 49]) and/or higher values of N, the study of entanglement properties [50–52], and the interplay between interactions and persistent currents [34]. Most importantly, the presented results open the door to the quantum simulation of strongly interacting multilayered solid-state devices coupled to external gauge fields by means of $SU(N)$ neutral atoms subject to Raman processes.

# Acknowledgements

The authors would like to thank A. Recati, L.F. Tocchio, and L. Livi for fruitful discussions. We acknowledge financial support from MIUR through the PRIN 2017 (Prot. 20172H2SC4 005) and PRIN 2020 (Prot. 2020JLZ52N 002) programs and Horizon 2020 through the ERC project FIRSTORM (Grant Agreement 692670) and ERC project TOPSIM (Grant Agreement 682629). L.D.R. acknowledges financial support from the U.S. Department of Energy, Office of Science, Basic Energy Sciences, Division of Materials Sciences and Engineering under Grant No. DE-SC0019469. We acknowledge financial support from PNRR MUR project PE0000023-NQSTI.

# A   Hartree-Fock method

In this section we provide some details about the static mean-field (Hartree-Fock) solution of the model (1). In the case $N = 2$, the only variational parameter included in our calculation is $s_{\mathbf{j}} = \langle d^{\dagger}_{\mathbf{j},1/2} d_{\mathbf{j},-1/2} \rangle$, which is also assumed to be uniform in the sample $s_{\mathbf{j}} = s$, and the Hamiltonian, written in momentum space, reads

$$\mathcal{H}_{\mathrm{MF}} = \sum_{\mathbf{k}} \psi^{\dagger}_{\mathbf{k}} \begin{pmatrix} \varepsilon_{1/2}(\mathbf{k}) & \Omega + Us \\ \Omega + Us & \varepsilon_{-1/2}(\mathbf{k}) \end{pmatrix} \psi_{\mathbf{k}} + UL^2 s^2 \,, \tag{A.1}$$

where we have introduced the spinor $\psi^{\dagger}_{\mathbf{k}} = (d^{\dagger}_{\mathbf{k},1/2}, d^{\dagger}_{\mathbf{k},-1/2})$ and the diagonal energy dispersion $\varepsilon_m(\mathbf{k}) = -2t \cos(\mathbf{k} \cdot \mathbf{u}_1 + m\gamma)$. The optimal value of $s$ can be obtained numerically by minimizing the Helmholtz free energy $F(s)$ of the system, which in the zero temperature case is reduced to the internal energy $E(s)$, obtained by filling the available energy states, starting from the lowest, with all the particles.

In the case $N > 2$ the scenario is much richer, since other mean-field parameters should be taken into account. For instance, the flavor-exchange processes are, in general, described by $N(N-1)/2$ variational parameters $s_{\mathbf{j};m,n} = \langle d^{\dagger}_{\mathbf{j},m} d_{\mathbf{j},n} \rangle$, with $m \neq n$. For $N = 3$, we have to include three flavor exchange parameters, which reduce to two by symmetry for the Raman tunneling scheme studied in this work: assuming again that they are uniform in the real dimensions and omitting the label $\mathbf{j}$, we have $s_{-1,0} = s_{0,1} := s$ and $s_{-1,1} := \tilde{s}$. Besides flavor exchange, another parameter should be introduced when more than two flavors are available, namely the imbalance in the population of different flavors $\nu_{\mathbf{j},m} = \langle d^{\dagger}_{\mathbf{j},m} d_{\mathbf{j},m} \rangle$. Such imbalance is forbidden when $N = 2$ by the reflection symmetry of the Hamiltonian mentioned in Sec. 2. Furthermore, if density fluctuations in the real dimensions are neglected, then by translation invariance one can omit the index $\mathbf{j}$ and the variational parameters satisfy the constraint $\sum_m \nu_m = 1$, which means that the independent parameters are $N - 1$. For $N = 3$ we would need two parameters to describe the flavor-population imbalance, but the specific hopping scheme studied here provides a reflection symmetry, offering another constraint $\nu_{-1} = \nu_1$ and limiting the number of independent parameters to one.

We can thus write a simplified variational ansatz by means of only three parameters: $\delta$, which measures the imbalance between the population of the outer and inner flavors ($\nu_0 = 1/3 + \delta$; $\nu_{-1} = \nu_1 = 1/3 - \delta/2$); $s$, which renormalizes the Raman matrix element ($\Omega_{\mathrm{eff}} \rightarrow \Omega + sU$); and $\tilde{s}$, which introduces an effective hopping between the external flavors with amplitude $-U\tilde{s}$. The mean field Hamiltonian, written in terms of $s$, $\tilde{s}$ and $\delta$ reads

$$\mathcal{H}_{\mathrm{MF}} = \sum_{\mathbf{k}} \psi^{\dagger}_{\mathbf{k}} \begin{pmatrix} \varepsilon_1(\mathbf{k}) + \frac{U\delta}{2} & \Omega + Us & -U\tilde{s} \\ \Omega + Us & \varepsilon_0(\mathbf{k}) - U\delta & \Omega + Us \\ -U\tilde{s} & \Omega + Us & \varepsilon_{-1}(\mathbf{k}) + \frac{U\delta}{2} \end{pmatrix} \psi_{\mathbf{k}} + UL^2 \left( 2s^2 + \tilde{s}^2 + \frac{3}{4} \delta^2 \right), \tag{A.2}$$

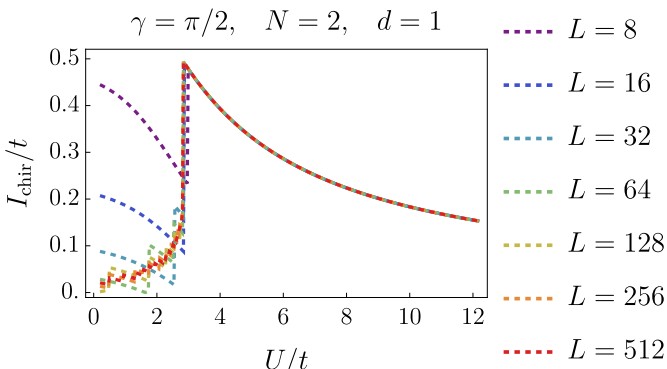

Figure 6: Finite-size scaling of the chiral current obtained by the Hartree-Fock method with $\Omega = 0.4\,t$. The plot shows the case $N = 2$, $d = 1$, which is representative of all the other cases studied in this work.

where now the spinor has three components $\Psi_{\mathbf{k}}^{\dagger} = (d_{\mathbf{k},1}^{\dagger}, d_{\mathbf{k},0}^{\dagger}, d_{\mathbf{k},-1}^{\dagger})$. Once again, the optimal values of the three variational parameters are obtained by minimizing the Helmholtz free energy, which is now a multivariate function $F(s, \tilde{s}, \delta)$, and reduces to the internal free energy $E(s, \tilde{s}, \delta)$ at zero temperature. After determining the optimal value of the variational parameters, one can then compute the associated expectation value of the chiral current (7). An example is illustrated in Fig. 6 for $N = 2$, $d = 1$, and different values of $L$. One can notice that, in the metallic phase, the functional dependence of the chiral current on $U/t$ gets increasingly regular upon increasing the value of $L$ because finite-size effects are washed away. Nevertheless, the important properties discussed in Sec. 3, i.e. the presence of a small chiral current in the metallic phase, a cusp-like maximum at the transition, and the $\sim (U/t)^{-1}$ behavior in the insulating phase, are always present, both in small-sized systems and approaching the thermodynamic limit. These observations should provide the correct interpretation of the ED results presented in Appendix C.

This approach can, in principle, be generalized to include other interesting effects, such as $SU(N)$-magnetic orderings along the real dimension, by making a reasonable variational ansatz relevant for the type of ordering under investigation.

## B Dynamical mean field theory

This section is devoted to a brief explanation of the DMFT approach to the problem investigated in the main text. This method amounts to map the lattice model into an effective impurity problem which is self-consistently determined. It is convenient to work in a grand canonical ensemble and to recast Hamiltonian (3) in the form:

$$\mathcal{H} = -t \sum_{\mathbf{ij}} \psi_{\mathbf{i}}^{\dagger} \cdot \mathbf{\Phi}_{\mathbf{ij}} \cdot \psi_{\mathbf{j}} + \sum_{\mathbf{j}} \psi_{\mathbf{j}}^{\dagger} \cdot M \cdot \psi_{\mathbf{j}} + \frac{U}{2} \sum_{\mathbf{j}} \psi_{\mathbf{j}}^{\dagger} \cdot \psi_{\mathbf{j}}(\psi_{\mathbf{j}}^{\dagger} \cdot \psi_{\mathbf{j}} - 1) - \mu \sum_{\mathbf{j}} \psi_{\mathbf{j}}^{\dagger} \cdot \psi_{\mathbf{j}}, \quad \text{(B.1)}$$

where we have introduced the $N$-dimensional spinor $\psi_{\mathbf{j}}^{\dagger} = (d_{\mathbf{j},\mathcal{I}}^{\dagger}, ..., d_{\mathbf{j},-\mathcal{I}}^{\dagger})$ (the real space counterpart of the spinor defined in Appendix A), and the following matrices:

$$[\mathbf{\Phi}_{\mathbf{ij}}]_{mm'} = \delta_{mm'}\delta_{\mathbf{j},\mathbf{i}\pm\mathbf{u}_1}e^{im\mathbf{\gamma}\cdot(\mathbf{j}-\mathbf{i})}, \qquad [M]_{mm'} = \Omega(\delta_{m',m+1} + \delta_{m',m-1}).$$

The chemical potential $\mu$ is adjusted to obtain the desired filling of one particle per site.

In order to have a manifestly translation-invariant model, we perform the unitary transformation $\psi_{\mathbf{i}} \rightarrow \mathcal{U} \cdot \psi_{\mathbf{i}}$, where $\mathcal{U}$ is a unitary matrix that diagonalizes $M$, i.e. such that

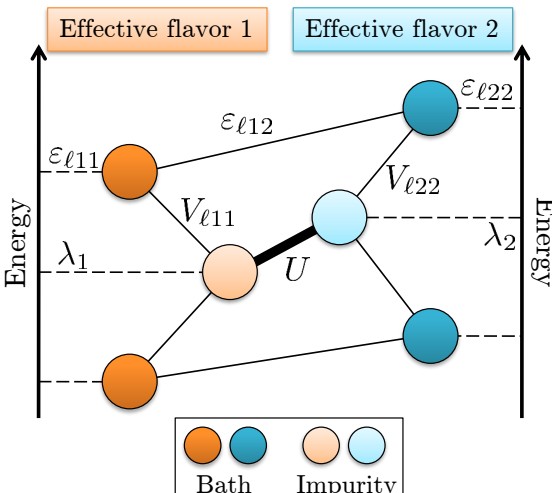

Figure 7: Sketch of the effective impurity problem used for DMFT in the system with $N = 2$. Each color represents a different internal state in the basis that diagonalizes the Raman matrix $M$ (effective flavor); darker circles represent bath sites, while lighter circles represent impurities. Each term in Hamiltonian (B.3) is represented by a line: solid thin lines represent tunnelings, dashed lines represent on-site energies and the solid thick line represents the Hubbard interaction.

$\mathcal{U} \cdot M \cdot \mathcal{U}^\dagger = \lambda$, where $\lambda = \text{diag}(\lambda_1, ..., \lambda_N)$ with $\lambda_i$ being the eigenvalues of $M$. In the new basis of "effective flavors", Hamiltonian (B.1) takes the form:

$$\mathcal{H} = -t \sum_{\mathbf{ij}} \psi_{\mathbf{i}}^\dagger \cdot \rho_{\mathbf{ij}} \cdot \psi_{\mathbf{j}} + \sum_{\mathbf{j}} \psi_{\mathbf{j}}^\dagger \cdot \lambda \cdot \psi_{\mathbf{j}} + \frac{U}{2} \sum_{\mathbf{j}} \psi_{\mathbf{j}}^\dagger \cdot \psi_{\mathbf{j}} (\psi_{\mathbf{j}}^\dagger \cdot \psi_{\mathbf{j}} - 1) - \mu \sum_{\mathbf{j}} \psi_{\mathbf{j}}^\dagger \cdot \psi_{\mathbf{j}}, \quad \text{(B.2)}$$

where $\rho_{\mathbf{ij}} = \mathcal{U} \cdot \Phi_{\mathbf{ij}} \cdot \mathcal{U}^\dagger$.

Following standard derivations of DMFT, we map the model onto the impurity model

$$\mathcal{H}_{\text{eff}} = \sum_{\ell=1}^{N_s} \phi_\ell^\dagger \cdot \epsilon_\ell \cdot \phi_\ell + \sum_{\ell=1}^{N_s} \left( \phi_\ell^\dagger \cdot V_\ell \cdot \psi + \text{h.c.} \right) + \psi^\dagger \cdot (\lambda - \mu) \cdot \psi + \frac{U}{2} \psi^\dagger \cdot \psi \, (\psi^\dagger \cdot \psi - 1), \quad \text{(B.3)}$$

which is schematically represented in Fig. 7. In this model $\psi_\alpha^\dagger$ creates a particle with effective flavor $\alpha = 1, ..., N$ in an impurity site, where particles experience the Hubbard interaction $\propto U$ and which is hybridized with a non-interacting bath, here parameterized by $N_s$ bath "sites" associated with creation operators $\phi_{\ell, \alpha}^\dagger$. Each site of the bath is coupled with the impurity via flavor-dependent tunneling terms encoded in the $N \times N$ matrices $V_\ell$ and it features flavor-dependent on-site energies, and local transitions between different flavors, both included in the $N \times N$ real symmetric matrices $\epsilon_\ell$.

The parameters defining $\epsilon_\ell$ and $V_\ell$ are determined self-consistently, by requiring the equivalence between the effective Green function of the impurity problem $G_{\text{eff}}(i\omega_n)$ and the local Green function of the lattice model $G(i\omega_n) = \frac{1}{L^d} \sum_{\mathbf{k}} G(\mathbf{k}, i\omega_n)$, where $i\omega_n$ are fermionic Matsubara frequencies. The latter condition can be recast as

$$\frac{1}{L^d} \sum_{\mathbf{k}} \left[ G_0^{-1}(\mathbf{k}, i\omega_n) - \Sigma(i\omega_n) \right]^{-1} = G_{\text{eff}}(i\omega_n), \quad \text{(B.4)}$$

where $G_0^{-1}(\mathbf{k}, i\omega_n) = i\omega_n + \mu + t \rho_{\mathbf{k}} - \lambda$, with $\rho_{\mathbf{k}} = \frac{1}{L^d} \sum_{\langle \mathbf{ij} \rangle} e^{i\mathbf{k} \cdot (\mathbf{j} - \mathbf{i})} \rho_{\mathbf{ij}}$, and $\Sigma(i\omega_n)$ is the impurity self-energy of the effective model defined in Eq. (B.3), that can be extracted from

the local Dyson equation:

$$\Sigma(i\omega_n) = G_{0,\text{eff}}^{-1}(i\omega_n) - G_{\text{eff}}^{-1}(i\omega_n), \tag{B.5}$$

with $G_{0,\text{eff}}(i\omega_n)$ being the non-interacting propagator of the impurity problem. We notice that, by construction, the self-energy of the system coincides with the self-energy of the associated impurity problem: therefore it does not depend on the crystalline momentum $\mathbf{k}$. This feature of the self-energy is exact only in the limit of infinite dimensions, while it represents an approximation in finite dimensions.

A DMFT solution amounts to solve iteratively the impurity model computing $\Sigma(i\omega_n)$ or equivalently $G(i\omega_n)$, imposing the self-consistency condition (B.4). Here we use an exact diagonalization solver based on the Lanczos method [31, 36, 53].

Finally, the converged self energy can be used to compute relevant observables, such as the chiral current. This boils down to compute expectation values of momentum-resolved density operators in the original basis of the physical flavors $\langle n_{\mathbf{k},m} \rangle$, where $m = -\mathcal{I}, ..., +\mathcal{I}$, which can be done by means of

$$\langle n_{\mathbf{k},m} \rangle = \lim_{\eta \to 0^+} \frac{1}{\beta} \sum_{\omega_n} \left[ \mathcal{U} \cdot G(\mathbf{k}, i\omega_n) \cdot \mathcal{U}^\dagger \right]_{mm} e^{-i\omega_n \eta}, \tag{B.6}$$

where $G(\mathbf{k}, i\omega_n) = (G_0^{-1}(\mathbf{k}, i\omega_n) - \Sigma(i\omega_n))^{-1}$ is the converged Green function of the system. The results presented in the main text have been obtained by fixing a finite number of bath sites: $N_s = 5$ for $N = 2$, and $N_s = 3$ for $N = 3$.

## C  Currents in the ladder-like system

In this section we present the exact solution of the system in $(1+1)$-dimensional clusters of small size, i.e. on a ladder geometry. On the one hand this enables us to study the fate of chiral currents in the presence of thermal fluctuations, which must be taken into account in experimental setups. On the other hand we can go beyond the translation-invariance requirement introduced to study the two-dimensional system, and study the space-resolved current patterns, which display a totally different behavior depending on the interaction strength between particles, resulting in the Vortex-Meissner transition.

### C.1  The role of temperature

To begin with, we investigate the robustness of our results with respect to temperature, which is a crucial aspect in the experimental realizations, that are limited to finite temperatures. Even though thermodynamic properties of SU(N)-Hubbard models [54, 55] have been investigated, less attention has been given to the impact of temperature on chiral currents in presence of artificial gauge fields.

DMFT can be used in principle to determine the thermal equilibrium state of the system, however our Lanczos-based solver is limited to low temperatures [36]. Hence we proceed with an exact diagonalization (ED) of small clusters which, at the same time, provides us with a benchmark of the DMFT results. However, since ED is limited to small cluster sizes, we consider only the case of $(1+1)$-dimensions (Fig. 8).

The $T = 0$ results clearly show a qualitative agreement with DMFT (top row of Fig. 3). This is a non-trivial result in the light of the different advantages and disadvantages of the two methods (DMFT works in the thermodynamic limit, but it neglects non-local spatial correlations, while ED is limited to small clusters), which strongly suggests that our results do not depend on the specific approximations inherent to the two methods. The main difference

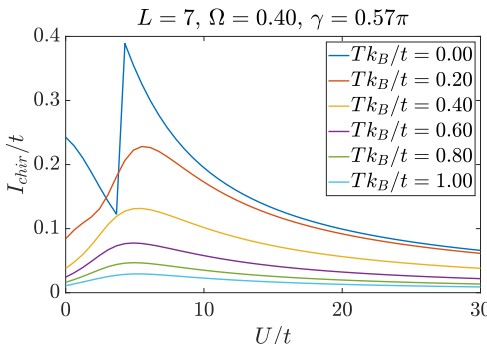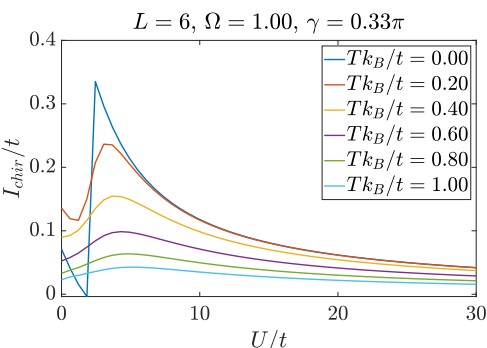

Figure 8: Chiral current as a function of $U/t$ and for different temperatures $T$. The presence of a maximum in the function $I_{\text{chir}}(U)$ is robust up to temperatures of order $\sim t/k_B$, and hence liable of experimental detection. The presence of a relatively large current at $U \ll t$ and $k_B T \ll t$ should be regarded as a finite-size effect, as we pointed out in Appendix A. Results have been obtained by means of the exact numerical diagonalization of Hamiltonian (1) in (1+1) dimensions with $L$ sites. Left (right) panel corresponds to $N = 2$ ($N = 3$).

emerging from these two methods is the quantitative value of the chiral current on the metallic phase, which is larger in systems of small size. This is a finite-size effect due to the presence of few quasiparticle states, that make a metal barely distinguishable from an insulator in a small cluster. The disruptive interference mechanism responsible for the current drop in the metal, described in Sec. 3, is thus not very effective in small systems, resulting in a large current. Nevertheless, it is worth observing that the chiral current behavior in the insulator, as well as the peak at the transition point is already captured in systems of small size.

Turning to the temperature dependency, we find that the peak of $I_{\text{chir}}$ is smeared by increasing temperature. Yet, the $T = 0$ picture qualitatively survives up to temperatures of the order of some tenths of $t/k_B$; which are, in fact, the typical operating conditions of state-of-the-art experimental platforms [12] (see also Sec. 6). We notice that the decrease of the chiral current as a function of temperature is not due to charge excitations across the Mott gap ($\sim U$), while it follows from the suppression of virtual hopping processes, i.e. the onset of flavor excitations. In Fig. 9, we show several thousands of low-lying eigenvalues $E_j$ of Hamiltonian (1) as a function of $U/t$, together with the double occupancy of each state $\langle \psi_j | \hat{D} | \psi_j \rangle$, where $\hat{D} = L^{-d} \sum_{\mathbf{i}} \sum_{m < m'} n_{\mathbf{i},m} n_{\mathbf{i},m'}$. For large values of $U/t$, two bundles can be identified, the lower (upper) one corresponding to states featuring zero (one) doublon-holon excitations (see Ref. [56] for a thorough discussion about the hierarchy of excitations in multicomponent fermionic systems). Since the upper band is activated only at rather high temperatures ($\sim U/k_B$), it is indeed clear that, in the strong-coupling regime, it cannot play any role in the suppression of the chiral current.

### C.2 The Vortex-Meissner transition

Once the qualitative agreement between DMFT and ED results has been established, one can use the latter to investigate the spatial configuration of observables. For this reason we consider the interacting $(1 + 1)$-dimensional system with open boundary conditions (OBC) along both the real and the synthetic directions, which introduces a difference between edge sites and bulk sites, and we investigate how the spatial current pattern is modified across the $U$-driven metal-insulator transition. We notice that DMFT can not be straightforwardly used with OBC because it requires translation invariance.

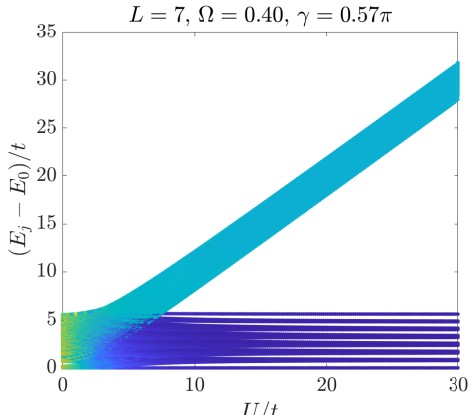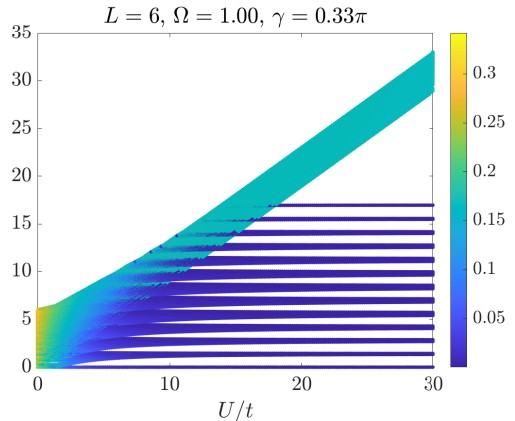

Figure 9: Eigenvalues of (1) as a function of $U/t$. The color code corresponds to the expectation value $\langle \psi_j | \hat{D} | \psi_j \rangle$ of the double-occupations operator $\hat{D}$. The horizontal dark blue bundles correspond to states featuring flavor excitations, while the lighter diagonal stripe represents states featuring a single doublon-holon excitation. States featuring multiple doublon-holon excitations or triple occupations are not displayed as they occur at even higher energies. The physical parameters correspond to those used in Fig. 8. Left (right) panel corresponds to $N = 2$ ($N = 3$) and include 500 (1000) energy levels.

In any stationary state we have, on every site $\mathbf{i}$, $\langle \dot{n}_{\mathbf{i},m} \rangle = 0$, where $\dot{n}_{\mathbf{i},m}$ is the time derivative of the number operator $n_{\mathbf{i},m} = d_{\mathbf{i},m}^{\dagger} d_{\mathbf{i},m}$. This means that the current flowing into each site $(\mathbf{i}, m)$ equals the current flowing out of it, a prescription which is equivalent to the Kirchhoff's law for the node. This statement implies that

$$\left\langle \sum_a \left( I_{\mathbf{i},\mathbf{i}-\mathbf{u}_a;m}^a + I_{\mathbf{i},\mathbf{i}+\mathbf{u}_a;m}^a \right) + I_{\mathbf{i};m,m+1} + I_{\mathbf{i};m,m-1} \right\rangle = 0 \,, \tag{C.1}$$

where

$$I_{\mathbf{i},\mathbf{i}\pm\mathbf{u}_a;m}^a = \mathbf{I}_{\mathbf{i},\mathbf{i}\pm\mathbf{u}_a;m} \cdot \mathbf{u}_a = -it \left( e^{-im\boldsymbol{\gamma}\cdot\mathbf{u}_a} d_{\mathbf{i},m}^{\dagger} d_{\mathbf{i}\pm\mathbf{u}_a,m} - \text{h.c.} \right) \,, \tag{C.2}$$

is the signed current along the real lattice bond from node $(\mathbf{i}; m)$ to node $(\mathbf{i}-\mathbf{u}_a; m)$, while

$$I_{\mathbf{i};m,m+1} = i\Omega \left( d_{\mathbf{i},m}^{\dagger} d_{\mathbf{i},m+1} - \text{h.c.} \right) \,, \tag{C.3}$$

is the signed current along the synthetic bond from node $(\mathbf{i}; m)$ to node $(\mathbf{i}; m+1)$.

The ED results are shown in Fig. 10, where we compare one calculation representative of the metallic region and one for the insulating region for $N = 3$ (the results for $N = 2$ are qualitatively similar). The currents are illustrated by arrows connecting nodes of the synthetic ladder, which are represented by circles, whose area reflects the average density $\langle n_{\mathbf{i},m} \rangle$. These results show that the metal-insulator transition is reflected in an interaction-driven transition between a vortex phase and a Meissner phase [57–59]. At weak-coupling (left panel) the currents along the synthetic dimension (vertical arrows) are non-vanishing and their magnitude and sign are site-dependent, leading to a vortex pattern where vortices of opposite charge are alternating in real space. On the other hand, in the strong-coupling regime (right panel), the currents in the synthetic dimension are zero everywhere except for the two outermost sites in the physical dimension. This result, together with the continuity equations (C.1), supports the fact that, in the insulating state, currents are expelled from all inner bonds and can circulate only along the outer boundary of the synthetic two-dimensional system, a circumstance

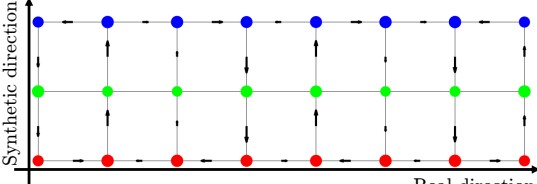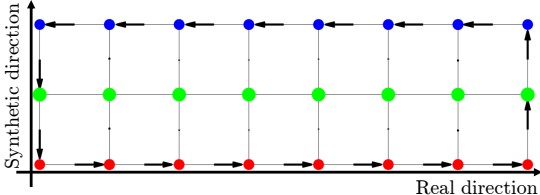

Figure 10: Current pattern in the metallic ($U/t = 2.0$, left panel) and in the insulating ($U/t = 8.5$, right panel) regimes. In the former case, vertical bonds are, in general, flown by a non-zero current resulting in multiple vortices; while in the latter case the current flows only along the edges of the ladder, resulting in a single-vortex structure, reminiscent of the Meissner effect in superconductors. Results are obtained by means of the exact numerical diagonalization of Hamiltonian (1) for a $(1 + 1)$-dimensional system, $N = 3$ flavors, and OBC. The currents flowing in each bond (black arrows) have been explicitly checked to satisfy continuity equation (C.1). Model parameters $\Omega/t = 0.5$, $\gamma = 2\pi/7$, and $L = 8$ have been used.

which is reminiscent of the Meissner effect in superconductors (see also Sec. 5 for an effective magnetic model accounting for this phenomenon).

We remark that, as evidenced from our ED results, in finite-size systems, vortex-like configurations of current patterns undergo structural changes upon increasing $U/t$. This is an effect of the (in)commensurability of the vortex typical size and the length of the system, which results in a functional dependence $I_{\mathrm{chir}}(U)$ characterized by a series of non-differentiable points, each of them corresponding to a re-arrangement of the vortex-like current pattern. These singularities constitute a finite-size effect, and get more rarefied and less pronounced upon increasing the system size.

According to the effective strong-coupling model discussed in Sec. 5, the chiral current (15) in the insulating phase is expected to be parallel to $\boldsymbol{\gamma}$. So, if $\boldsymbol{\gamma}$ is taken to be parallel to the $x$-axis, i.e. $\boldsymbol{\gamma} = \gamma\mathbf{u}_x$, than the Meissner-like current pattern illustrated in the right panel of Fig. 10 is expected to hold at each section $xm$ or, in other words, to be replicated along the $y$-axis.

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
