# Peer review of "Enhancement of chiral edge currents in ($d$+1)-dimensional atomic Mott-band hybrid insulators"

_SciPost Physics, doi:SciPost Phys. 14, 048 (2023)_

## Round 1 · Referee Report · Anonymous (Referee 1) · 2022-9-22

Report
In the paper, a multi-flavor model in one and two dimensions is discussed. The SU(N) symmetry (with N=2 and 3) is broken by means of Raman processes that can be regarded as tunneling events between internal degrees of freedom, leading to an extra dimension in the problem. A repulsive interaction term is added to penalize double and multiple occupations of each site and the authors work at integer filling in order to investigate the Mott physics.
The paper is clearly written and the results are in general well supported by considering different regimes of the parameters as well as different techniques. The most important outcomes are relative to the study of chiral currents, that have a maximum at the metal-insulator transition and are present also in the insulating phase. Interestingly, the authors show also that the insulating phase has a hybrid Mott and band nature that is not so common in the context of strongly correlated systems. The results are presented in view of a possible experimental realization; even if the the number of flavors N is larger in experiments, the fact that the results are valid both for N=2 and for N=3 supports the idea that they can be valid also for a larger value of N. I also appreciate Appendix A that clarifies a non trivial application of the Hartree-Fock approach.
From these considerations, I think that the paper has a high enough level of originality, clarity and solidity to be accepted. Nevertheless, some points need to be discussed before publication, as listed below:
-) It is not clear to me why the chiral currents have to be even larger in the insulating phase than in the metallic phase. Can the authors comment more on this point?
-) The current pattern is studied, via exact diagonalization, only in one dimension. Can the authors describe almost qualitatively how they are expected to behave in two dimensions? Related to this point, I do not understand much the reference to the Meissner effect in superconductors when we are in the insulating phase. Which should be the relation between the presented model and a superconductor?
-) It is interesting to see how the insulating phase mixes band and Mott nature in it. Still, it is unclear to me why the limit of large U should not be dominated only by the Coulomb repulsive effects. I would expect that when U is the dominating energy scale, I should see only the Mott physics. Can the authors comment more on this?
-) In Sec. 7, the fact that the chiral current for N=3 is formally equal to the one for N=2 is not obvious to me. Can the authors include more steps in the derivation of Eq. (23)?
In addition, I have the following two minor points:
-) The yellow lines in Fig. 6 are not always well visible. If possible, I would suggest the authors to improve the presentation.
-) At the end of Sec. 6, the authors refer to Sec. 2, but I think that they refer to Sec. 3. Is this correct?
Author: Andrea Richaud on 2022-12-13 [id 3129]
(in reply to Report 1 on 2022-09-22)We thank the Referee for her/his careful, constructive and positive report. We also thank her/him for providing further useful comments and suggestions for improving it. We reply point-by-point in the attached pdf document (please see comments to the resubmission).
Author: Andrea Richaud on 2022-12-12 [id 3128]
(in reply to Report 2 by Andreas Haller on 2022-10-07)We thank the Referee for his careful, constructive and positive report. We also thank him for providing further useful comments and suggestions for improving it. We reply point-by-point in the attached pdf document (please see comments to the resubmission).
Attachment:
22_12_12_-_Reply_to_Referees.pdf

---

## Round 1 · Referee Report · Andreas Haller (Referee 2) · 2022-10-7

Report
The authors study the chiral edge currents of quasi-3D bilayer systems subject to local interactions along the synthetic dimension.
I personally find the subject very interesting and timely, and their findings are presented in an appealing way. Although the counterintuitive or ''anomalous'' behavior of currents upon repulsive interactions and their associated susceptibilities have already been addressed in the literature of (quasi) 1d ladder systems, their higher-dimensional analogs are far from being fully understood.
To my knowledge, this is the first study that presents and compares numerical and analytical results on the chiral currents of various ground states in these quasi-2d systems. The authors here use a mean-field approach, predictions from a strong coupling limit, and numerical DMFT simulations which all agree qualitatively (and quantitatively only in some limits). Therefore, their presented numerical results are thoroughly proof-checked and trustworthy.
The authors further argue by ED results that 1d ladder systems can be used to understand the phases qualitatively. I find this connection very intriguing, but similarly, I have to say that this is perhaps the biggest weak point of this work. In my perspective, it is misleading to analyze the physics of 1d models and, based on macroscopic observables (the average chiral current), suggest that microscopic details (nodal current flows) are similar in a 2d lattice without an attempt to study small 2d clusters, or present the same quantities using a full-fledged 2d method capable of doing so. I suggest two possibilities to improve the manuscript in this respect. The authors could extend this section by including a study of small 2d clusters with an appropriate technique of their choice (e.g. density group renormalization group or auxiliary field quantum Monte-Carlo frameworks) which would not only bridge the gap between 1d and 2d, but significantly strengthen the concluding arguments as well. As an alternative, the authors could remove the misleading implications, combine the short sections 4 and 5 and move them to a dedicated appendix.
Requested changes
Besides the major point mentioned in the main report, here is my list of suggestions
1) Fig 1. (c), add missing hopping arrows
2) First term of Eq. (1) has a missing 1/2 in front (bonds are accounted for twice in the sum).
3) Perhaps mention I_chir = \braket{\partial H/\partial\gamma} around Eq. (6), or is this equation only true in 1d?
4) Below Fig. 2: ''... the flavor polarization ... changes'' better: decreases?
5) Above Fig. 3: please mention existing literature in 1d concerning anomalous Drude weight effects.
6) Below Fig. 4: "The T=0 results clearly show a qualitative agreement." Please comment on the strong v-shaped kink at U=0, which is clearly absent in your 2d estimates.
7) Section 5: see main report. Is it possible to artificially double the impurity unit cell in order to compute current correlations with DMFT? An alternative to bridge the gap between 1d and 2d would be the utilization of DMRG frameworks such as iTensors or TenPy, or auxiliary field quantum Monte-Carlo frameworks such als ALF, which have similar models already implemented.
8) Fig. 6: perhaps use the same aspect ratio between x and y links, otherwise it is impossible to guess by eye that the Kirchhoff law is satisfied
9) General comment on the mean-field approach: the currents in Fig. 6 left panel are clearly site-dependent. Provided currents in the 2d case behave similarly, why is this site dependence neglected on the mean-field (and DMFT) level?
10) Why ''synthetic ladder'' with quotation marks?
11) Could the authors comment on my following question: Is the chiral current in these models related to any topological phase, or predecessors thereof, similarly to the scenario for quasi 1d ladders threaded with flux, which are related to quantum Hall phases via the coupled wire formalism?
12) If the conjecture that 1d and 2d insulating phases are qualitatively similar is correct, the chiral current is apparently carried by the exterior synthetic sheets. Could you confirm this by presenting/producing also DMFT results for N=4?
13) Can you compute and present the current between the synthetic sheets with DMFT? This would further establish a similarity to Fig. 6.

---

## Round 2 · Referee Report · Anonymous (Referee 3) · 2022-12-29

Report

The authors replied to my comments and to the comments of the second referee in a very precise and pertinent way, I have rarely seen such an in-depth reply! For this reason, the paper is definitely ready to be accepted for publication. I also have to say that I appreciate the new structure of the paper since I agree with the main comments of the second referee.

---

## Round 2 · Referee Report · Andreas Haller (Referee 2) · 2023-1-12

Report

The authors have submitted a thorough reply together with an improved version of the manuscript, addressing all points raised in the last round. I appreciate the new structure of the paper and that many suggestions have found their way into the final manuscript.

With these changes, I have no further comments or remarks to add.

Requested changes

-

---

## Round 2 · Author Response

Dear Editor,

thanks for sending us the reports of two Referees. We are grateful to the Referees for their careful, positive and constructive reports that helped us improving our manuscript.
We have indeed revised the manuscript complying with the requests of both Referees. Our detailed point-by-point response to the Referees has been uploaded.

In view of the already positive report of both Referees, and in view of the improvements that we have introduced in order to fully comply with the requests of both Referees, we are confident that this new version of our paper can be accepted for publication on SciPost Physics.

Best regards,

The authors

---

## Round 2 · List of Changes

A detailed list of changes has been attached as a pdf file (please see Comment to the Resubmission).

---

## Editorial Decision

published